# Age-Dependent Decline of NAD^+^—Universal Truth or Confounded Consensus?

**DOI:** 10.3390/nu14010101

**Published:** 2021-12-27

**Authors:** Augusto Peluso, Mads V. Damgaard, Marcelo A. S. Mori, Jonas T. Treebak

**Affiliations:** 1Novo Nordisk Foundation Center for Basic Metabolic Research, Faculty of Health and Medical Sciences, University of Copenhagen, DK 2200 Copenhagen, Denmark; augusto.peluso@sund.ku.dk (A.P.); damgaard@sund.ku.dk (M.V.D.); 2Department of Biochemistry and Tissue Biology, Institute of Biology, University of Campinas, São Paulo 13083-862, Brazil; morima@unicamp.br; 3Obesity and Comorbidities Research Center, University of Campinas, São Paulo 13083-862, Brazil; 4Experimental Medicine Research Cluster, University of Campinas, São Paulo 13083-862, Brazil

**Keywords:** NAD^+^, aging, yeast, *C. elegans*, mouse, rat, monkey, human

## Abstract

Nicotinamide adenine dinucleotide (NAD^+^) is an essential molecule involved in various metabolic reactions, acting as an electron donor in the electron transport chain and as a co-factor for NAD^+^-dependent enzymes. In the early 2000s, reports that NAD^+^ declines with aging introduced the notion that NAD^+^ metabolism is globally and progressively impaired with age. Since then, NAD^+^ became an attractive target for potential pharmacological therapies aiming to increase NAD^+^ levels to promote vitality and protect against age-related diseases. This review summarizes and discusses a collection of studies that report the levels of NAD^+^ with aging in different species (i.e., yeast, *C. elegans*, rat, mouse, monkey, and human), to determine whether the notion that overall NAD^+^ levels decrease with aging stands true. We find that, despite systematic claims of overall changes in NAD^+^ levels with aging, the evidence to support such claims is very limited and often restricted to a single tissue or cell type. This is particularly true in humans, where the development of NAD^+^ levels during aging is still poorly characterized. There is a need for much larger, preferably longitudinal, studies to assess how NAD^+^ levels develop with aging in various tissues. This will strengthen our conclusions on NAD metabolism during aging and should provide a foundation for better pharmacological targeting of relevant tissues.

## 1. Introduction

Nicotinamide adenine dinucleotide (NAD^+^) can be generated from tryptophan or micronutrient precursors from the Vitamin B_3_ family, which consist of nicotinamide (NAM), nicotinic acid (NA), and nicotinamide riboside (NR). NAD^+^ precursors are micronutrients naturally found in the diet and can be obtained from different vegetal and animal food sources, found in high levels in, for example, cucumber, cabbage, soybeans, broccoli, avocado, tomato, whole wheat, yeast, eggs, milk, meat, and liver.

NAD^+^ is an important cofactor for adenosine triphosphate (ATP) production in glycolysis and oxidative phosphorylation as well as in cellular redox reactions by oxidoreductase enzymes [1]. NAD^+^ also functions as an essential co-substrate in pathways that regulate a wide variety of cellular processes such as DNA repair [2], cellular senescence [3,4], and mitochondrial respiratory function [5]. Consistent with its pleiotropic roles, NAD^+^ is a tightly regulated molecule, the levels of which are maintained through a fine balance between producing and consuming processes [6].

NAD^+^ is synthesized through distinct pathways in mammalian cells, including the following pathways (Figure 1): the kynurenine pathway, the Preiss–Handler pathway, and the salvage pathway. The kynurenine pathway involves the conversion of tryptophan into 2-amino-3-carboxymuconate semialdehyde (ACMS) via indoleamine-2,3-dioxygenase/tryptophan-2,3-dioxygenase (IDO/TDO). Through a spontaneous reaction, ACMS can be converted into quinolinic acid (QA) and subsequently into nicotinic acid mononucleotide (NAMN) via quinolinic acid phosphoribosyltransferase (QPRT) [7]. This NAD^+^ synthesis pathway is particularly important in the liver [8] and in macrophages [9].

The Preiss–Handler pathway involves the formation of NAMN from NA via nicotinic acid phosphoribosyltransferase (NAPRT). The NAMN resulting from both the kynurenine and the Preiss–Handler pathway, as well as from the conversion of nicotinic acid riboside (NAR) by NR kinases [10], is converted into nicotinic acid adenine dinucleotide (NAAD) via the nicotinamide mononucleotide adenylyl transferases (NMNATs) and finally to NAD^+^ via NAD synthetase (NADSYN1) [11,12,13,14].

The salvage pathway involves the regeneration of NAD^+^ by salvaging NAM after NAD^+^ consumption as described below. NAM is converted to nicotinamide mononucleotide (NMN) via nicotinamide phosphoribosyltransferase (NAMPT), which is the rate-limiting enzyme in the salvage pathway [15]. Additionally, nicotinamide riboside (NR) is also converted to NMN via NR kinases (NRK) [16], and NAD^+^ is subsequently produced from NMN via NMNATs [11,17,18].

NAD^+^ consumption results from the normal function of a variety of enzymes such as sirtuins, ADP-ribosyltransferases (ARTs), ADP-ribosyl cyclase/cyclic ADP-ribose hydrolase 1 (CD38), and the NAD^+^ hydrolase SARM1. In common for all these NAD^+^-consumers is the breakdown of NAD^+^ into ADP-ribose and NAM. Sirtuins are deacylases that participate in a series of cellular reactions, including cell metabolism, inflammation, apoptosis, and senescence [19,20]. ARTs are involved in cellular repair processes, such as DNA repair, but they also regulate telomere length, transcriptional processes, and the cell cycle [21,22,23]. CD38 is an NADase enzyme involved in cell adhesion, signal transduction and calcium signaling and plays a role in many cellular processes associated with aging and age-related chronic diseases [24] such as obesity [25,26], metabolic syndrome [27] and cancer [28,29]. Finally, SARM1 has pro-neurodegenerative properties, and while its function is not fully understood, it likely involves calcium mobilization through the generation of cyclic ADP-ribose (cADPR) [30,31,32].

Although the pathways controlling cellular NAD^+^ content are tightly regulated, decreased levels of intracellular NAD^+^, as well as in the NAD^+^/NADH ratio, have been observed during aging and aging-related pathophysiological conditions, such as metabolic diseases [13,33], oxidative stress [34], mitochondrial dysfunction [35], inflammation, and DNA damage [36]. These NAD^+^ imbalances occur either due to the downregulation of NAD^+^ biosynthesis, increased activity of NAD^+^ consuming enzymes or both [12,37]. Several studies have focused on elucidating the mechanisms of NAD^+^ maintenance in order to unlock its potential benefits for maintaining cell function and, consequently, preserving organismal vitality during aging [11,12,38]. By doing so, these studies aimed to formulate interventions capable of preventing the development of age-related diseases associated with lower levels of NAD^+^, such as diabetes [34,39,40,41,42], metabolic syndrome [13], neurodegenerative diseases [4,35,43,44,45], and cancer [2,18,46].

While there is certainly evidence of a decline in NAD^+^ levels in age-related diseases in both humans and animal models, reductions in NAD^+^ as part of physiological aging are also commonly being touted as a universal truth for all tissues in all organisms. This review will discuss the current literature linking NAD^+^ levels to aging across multiple tissues and species, including humans, to possibly challenge the apparent consensus in the field that NAD^+^ levels universally decline with age (Figure 2).

## 2. Relationship between NAD^+^ Levels and Aging across Species

### 2.1. Non-Mammalian Species

#### 2.1.1. Yeast

Although budding yeast can be propagated in culture indefinitely, the replicative lifespan of a single yeast cell, which is defined as the number of buds produced before death, rarely reaches beyond forty [47]. Replicative aging in yeast is thought to resemble features of the aging process of other asymmetrically dividing eukaryote cells, and this property has made it a useful model to study mechanisms of aging. Another method of studying aging in yeast is through the assessment of chronological lifespan, defined as the time that a single non-proliferating cell survives after the diauxic shift [48]. The diauxic shift is reached when yeast cells switch from glucose fermentation to ethanol respiration and, in that process, most cells stop budding. The cells ultimately reach a stationary phase in which they are still viable but do not proliferate.

Several studies have shown that manipulation of the genes and/or proteins associated with enzymatic production or consumption of NAD^+^ can regulate both replicative and chronological yeast life span [49,50,51,52,53,54]. However, whether NAD^+^ levels decline with age is questionable. A study comparing *S. cerevisiae* of replicative age 0–1 and 7–8, respectively, with regard to replicative aging, did not find any significant differences in NAD^+^ levels, although we note a visual tendency towards a decrease with aging [55]. Similarly, a study focusing on the kynurenine pathway showed that the levels of combined NAD^+^ and NADH remained unchanged when comparing yeast at replicative age 0 and replicative age 16 [56]. However, in both studies, the replicative ages investigated (7–8 and 16, respectively) are not necessarily old, as the average number of cell divisions of yeast is around 20–25 [48]. This is supported by the survival curve of the latter study, which shows no obvious decline in survival at these replicative ages [56]. Thus, the comparisons made in these studies do not reflect aging across its whole life spectrum.

When levels of NAD^+^ were studied in continuously dividing cultures of *S. cerevisiae*, a significant decline of NAD^+^ levels was observed when high optical densities (OD) were reached [50]. However, it is noteworthy that daughter (and granddaughter) cells of lower replicative age were present in the high-OD culture and cell density could conceivably affect NAD^+^ levels irrespective of replicative age. It is therefore not possible to determine a causal relationship between replicative age and NAD^+^ levels in this type of experiment.

To our knowledge, only a single study has investigated NAD^+^ in the context of chronological aging. The results demonstrated that NAD^+^ levels markedly declined in the three days following the diauxic shift [57]. During this period, yeast cells cement their senescent status by ceasing budding activity, downregulating metabolic pathways and upregulating stress-resistance pathways [48]. Thus, in yeast cells, NAD^+^ reduction may represent an integral part of the metabolic shift associated with senescence. Whether this is a hallmark of aging *per se*, or simply part of the shift from an active metabolic state to a less active metabolic state is unclear.

Taken together, there is seemingly no direct evidence of a connection between aging and NAD^+^ level decline in yeast cells. Future studies should consider both replicative and chronological models of aging in yeast and assess the whole spectrum of yeast lifespan to fully determine the role of NAD^+^ in yeast aging.

#### 2.1.2. Caenorhabditis Elegans

*C. elegans* are free-living nematodes with a lifespan of approximately 18 to 20 days. With a genetic homology of up to 80% compared to humans, *C. elegans* are easily cultivated in the lab, genetically amenable, and therefore commonly used for aging studies. Remarkably, only two studies have reported on NAD^+^ levels in this context. NAD^+^ levels were found to be reduced in aged *C. elegans* (day 17 and day 8 in [58,59], respectively) compared to young controls (day 1). These studies indicated an association between NAD^+^ and lifespan, and many subsequent studies in *C. elegans* have focused on the role of NAD^+^ consuming enzymes or supplementation with NAD^+^ precursors to affect longevity [59,60,61]. Although it is beyond the scope of the present review to discuss these findings, information from these studies have provided a valuable foundation for aging-related studies in other model organisms.

#### 2.1.3. Drosophila Melanogaster

Drosophila melanogaster is a widely used model to study aging and aging-associated disease mechanisms [62,63], also in relation to NAD^+^ metabolism and biosynthesis [64,65]. However, we have not been able to identify any papers that address whether levels of NAD^+^ decline with age in this model. This is somewhat surprising given the ease with which such an experiment could be performed. On the other hand, if only whole fly NAD^+^ levels are measured, the relevance of such data in relation to understanding the complexity of specific aging mechanisms may be questioned.

### 2.2. Rodents

#### 2.2.1. Rats

Rats have not been studied extensively with regard to NAD^+^ levels during aging. In one study, researchers compared female Wistar rats of different ages. They found that NAD^+^ was reduced in liver, heart, kidney and lung of 24-month-old rats, compared to younger controls (3 and 12 months old, respectively) [66]. A follow-up study with a similar design revealed the same pattern of NAD^+^ decline in four different brain regions: hippocampus, cortex, cerebellum and brainstem [67]. In both studies, the reduction in NAD^+^ was accompanied by an increase in NADH in all the mentioned tissues and as a result, the NAD^+^/NADH-ratio was skewed in the oldest group. These observations were attributed to an increase in Poly(ADP-ribose)polymerase (PARP) activity. Similarly, isolated mesenchymal stem cells from young (1–2 months of age) and old (15–18 months of age) male Sprague Dawley rats also exhibited a reduction in NAD^+^ levels with age, which was suggested to be due to a reduction in NAMPT levels [68]. It is noteworthy, however, that the mesenchymal stem cells were kept in culture prior to the assessment, which could, in principle, affect the outcomes. The same group later demonstrated that induced cellular senescence of rat mesenchymal stem cells also resulted in a reduction in NAD^+^, again associated with NAMPT reduction [69]. Collectively, our understanding on changes in NAD^+^ levels in aging in rats is limited to only a few tissues or cell types, and the information appears to have derived from only two laboratories.

#### 2.2.2. Mice

Mice have been studied more extensively, but there are clear discrepancies between the studies. In one study, comparing young (3–6 months of age) to aged (25–31 months of age) mice, NAD^+^ levels were reduced in pancreas, white adipose tissue and an undisclosed type of skeletal muscle [40]. Interestingly, the aged group exhibited quite a large variance in liver NAD^+^ levels, as both the lowest and highest NAD^+^ levels were found in this group. As a result, liver NAD^+^ levels did not change significantly with age in this study. These results were supported by the fact that NAMPT levels were unaffected with age in the liver, while adipose tissue NAMPT levels were clearly reduced with age. Skeletal muscle NAMPT abundance showed the same pattern, although it did not reach statistical significance, and pancreas was apparently not tested for NAMPT content. The observations on liver NAD^+^ levels are further supported by two studies from our group. The first study demonstrated that both chow and high fat-diet fed mice at the age of 48 weeks (~12 months) did not have reduced NAD^+^ levels compared with 6- or 12-week-old mice [70]. Intriguingly, this study showed increased NAD^+^ levels in livers of mice terminated at 24 weeks (~6 months). Although we cannot explain this finding, it should be mentioned that we consider animals in this study to have been middle-aged rather than old. In the second study, we observed an age-dependent increase in liver NAD^+^ levels when 55- and 110-week-old mice were compared to younger animals [71].

In contrast, others have reported a reduction in NAD^+^ levels in liver, spleen, adipose tissue, and an undisclosed type of skeletal muscle from mice between the ages of 5 and 32 months of age [26]. These observations of liver and skeletal muscle NAD^+^ levels were supported by another study, which reported a reduction in NAD^+^ in both tissues when comparing 6- to 24-month-old mice [58].

NAD^+^ levels in the gastrocnemius muscle have been reported to decrease with age in two studies. In the first study, 22- and 30-month-old mice were compared to a 6-month-old group, and gastrocnemius NAD^+^ levels were significantly reduced in both older groups compared to the young controls [72]. In the second study, a comparison was made between 4- and 24-month-old mice and showed the same pattern [73]. Moreover, in this latter study, the decreased NAD^+^ levels were associated with a reduction in NAMPT levels, and they were rescued in animals overexpressing NAMPT in muscle. Interestingly, NAD^+^ from freshly isolated muscle stem cells from mice were slightly lower in cells from old (i.e., 22–24-month-old) compared to young (1-month-old) mice [5]. Thus, while the effect of age on mouse liver NAD^+^ levels differs between studies, levels of NAD^+^ in skeletal muscle seem to be consistently lower in aged animals.

A gradual decline in NAD^+^ with age was demonstrated in the hippocampus of the brain [74]. Specifically, NAD^+^ levels were reduced stepwise for ages between 1 and 3–4 months as well as between 3–4 months and 6 months of age. However, NAD^+^ levels did not significantly change between 6 and 10–12 months of age. Thus, questions arise as to whether the gradual NAD^+^ reduction is relevant for actual aging, or whether it is important for changes in neurological functions during progression to middle age. A reduction in hippocampal NAMPT between 6- and 18-month-old mice was demonstrated, but data on NAD^+^ levels from this latest timepoint were not provided. Nevertheless, the conclusions were supported by a separate comparison of hippocampi from 2-, 7- and 19-month-old mice, which revealed that NAD^+^ was reduced between the youngest and the oldest group, while the middle-aged group was not found to be different from either [75]. Further support for an age-dependent decline in NADH and nicotinamide adenine dinucleotide phosphate (NADPH) in the hippocampal area shows that the relative levels of these metabolites are significantly decreased in the dentate gyrus when comparing 10-week-old mice to both 20- and 30-week-old mice [76].

Whole-brain NAD^+^ were not lower in 8- vs. 2-month-old mice [77]. If anything, according to our visual assessment, there was an increase in total brain NAD^+^ during this period of reaching middle age, but this was not tested statistically. The effect of aging on cerebellum NAD^+^ levels has also been indirectly assessed by quantifying NAD^+^ levels in cerebella from 4- and 16-month-old mice as part of a larger setup [78]. The two groups were not compared statistically, but we note a slight tendency for lower NAD^+^ with age in this setting. The same study also revealed a reduction in the cerebellum NAD^+^/NADH ratio in a mouse model of accelerated aging, though the relevance of this to physiological aging is questionable.

Isolated fibroblasts from the tail-tip of old (22-month-old) and young (2-month-old) mice have also been compared. In this case, samples were separated into mitochondrial, nuclear and cytosolic fractions. In all three cellular compartments, NAD^+^ levels were reduced in the cells from the older group, and this was associated with a reduction in genes linked to NAD^+^ synthesis, i.e., nicotinamide nucleotide adenylyltransferases 1–3 (NMNAT1-3) and nicotinamide nucleotide transhydrogenase (NNT) [79]. It should be noted that the cells were cultured for 8 days prior to any measurements. Similarly, a study focusing on NAD^+^ synthesis in primary mouse peritoneal macrophages found lower levels of NAD^+^ in primary cells from old mice (16–20 months of age) compared to young mice (3 months of age), and linked this to a reduction in the activity of the kynurenine pathway [9].

Lastly, the most extensive study to date on the development of NAD^+^ levels with aging, utilized isotope tracing and mass spectrometry to investigate NAD^+^ metabolism in several tissues [80]. It showed that aged (25-month-old compared to 3-month-old) mice had a decrease in NAD^+^ in only 10 out of 21 tested tissues. In brown and two different white adipose tissues (retroperitoneal and inguinal), as well as in the jejunum, they observed a ~40–50% NAD^+^ reduction with age, whereas in gastrocnemius, soleus, quadriceps, liver, kidney, and descending colon NAD^+^ levels were reduced by ~10–20%. The NAD^+^ levels in the remaining 11 tissues including heart, brain, spleen, lung, pancreas, gonadal white adipose tissue, and multiple parts of the gastrointestinal tract were found to be unchanged by aging. Of note, the assessment of pancreas and spleen NAD^+^ levels disagrees with previous studies in mice. Similarly, this assessment of aging heart and lung NAD^+^ levels disagrees with the previously mentioned rat study, though they both observe an NAD^+^ reduction in the kidneys. The discrepancies between these two studies likely result from species differences in NAD^+^ regulation. Nevertheless, it is clear from this study alone that the apparent consensus that NAD^+^ levels universally decline with aging is inaccurate. Less than half of the tested tissues displayed significant changes in NAD^+^ levels with age and a large subset of these was only slightly affected.

In summary, despite reliable evidence for a decline in NAD^+^ with aging in skeletal muscles, some adipose tissues, and hippocampal areas of the brain, there are conflicting reports on liver, which may suggest that a factor other than age may be important, and should be further elucidated. In addition, skeletal muscle, adipose tissue, brain and liver are the most investigated tissues, whereas effects of age on NAD^+^ levels in heart, lung, spleen, pancreas, and intestine remain relatively uninvestigated.

### 2.3. Primates

#### 2.3.1. Monkeys

Rhesus monkeys (*Macaca mulatta*) are commonly used in aging research due to their social and physiological similarities to humans [81]. No studies appear to have reported on levels of NAD^+^ with aging in monkeys. However, one study in rhesus monkeys showed 2.5- and 2-fold increases in NADH levels in vastus lateralis muscle of middle-aged (15–16 years of age) and older (28–32 years of age) animals, respectively, compared to young controls (6–9 years of age). Moreover, the NAD^+^/NADH ratio was ~60% lower in middle-aged animals compared to young controls [82]. Although NAD^+^ levels were not reported in this study, they can easily be calculated from the available data by multiplying the NAD^+^/NADH ratio with NADH levels. This reveals that NAD^+^ levels are roughly doubled in old monkeys compared with young ones, and that the middle-aged monkeys are in-between. It is unclear why the authors did not include data on NAD^+^ in the paper, but they certainly contrast the data from mouse skeletal muscle mentioned above. A second study conducted in both male and female marmosets (Callithrix jacchus) analyzed the blood plasma metabolome of animals between 1–17 years of age [83]. Despite the lack of direct data on NAD^+^ levels, the study showed an age-dependent increase in metabolites belonging to the Vitamin B_3_ family, of which nicotinic acid and nicotinamide were specifically highlighted, but no absolute concentrations were provided. Collectively, the available data from monkeys are scarce and more studies are required to gauge the usefulness of these model organisms in studying NAD^+^ metabolism in relation to human health.

#### 2.3.2. Humans

In a similar fashion to the model organisms described above, a consensus regarding NAD^+^ with aging appears to have been reached for humans. However, the actual data are less definitive. The first indication of age-related NAD^+^ decline in humans was reported approximately a decade ago in a study that investigated the connection between oxidative stress and NAD^+^ metabolism in human skin samples from the pelvic region [84]. This study showed that NAD^+^ levels correlated negatively with increasing age in both males and females. This was even found to be true when excluding data from male newborns, who notably presented comparatively high NAD^+^ levels. In addition, the participants were separated into the following discrete age groups: newborns (0–1 years, *n =* 8), young adults (30–50 years, *n =* 12), middle age (51–70 years, *n =* 23) and elderly (>71 years, *n =* 5). Our calculations based on the reported data revealed a ~68% reduction in skin NAD^+^ between newborns and young adults, as well as a further ~60% reduction between young adults and middle age. However, there was no sign of further decline when progressing from middle age to aged, which could mean either that a plateau was reached or that the sample size for the elderly group was a limiting factor. Interestingly, newborns appear to have exceptionally high skin NAD^+^ levels [84]. If this is a common feature of tissues from newborns, it might be explained by the quantities of NAD^+^ precursors in mammalian, including human, milk [85], as well as the undeveloped microbiota at this stage. The microbiota metabolizes and utilizes NAD^+^ precursors [86,87], but before the microbiota is established, it is reasonable to assume that a greater number of NAD^+^ precursors is absorbed by the host. We also know, from animal studies, that NAD^+^ is especially important during early development [88], so this interplay between natural NAD^+^ precursor supplementation as well as its reduced breakdown in the gut may be important for early stage development.

In a later study, it was found that in a gender-mixed cohort and a female sub-cohort, the amount of total NAD(H) in cerebrospinal fluid (CSF) was ~14% lower in the older group (>45 years, *n =* 36) compared to the younger group (≤45 years, *n =* 34) [89]. It should be noted that this mixed cohort had a severe underrepresentation of young males, and thus the comparison describes results between younger females and a mix of older people of both sexes. To their credit, the authors called into question whether this modest reduction would have a biochemical impact on cell metabolism and suggested a need for follow-up studies. The authors also reported a correlation between plasma and CSF NAD(H) levels in 38 of the 70 participants where they had both types of samples. However, they were unable to demonstrate a direct effect of age on plasma NAD(H) levels and unfortunately did not disclose the age and sex of the 38 participants from which plasma was acquired. The same study also did not report on correlations between age and either CSF or plasma NAD(H) levels, although this does not exclude the possibility that such correlations exist. Importantly, the reported plasma NAD(H) levels of ~360 µg/mL are very high compared to the literature [90,91,92]. This is, in part, caused by the measuring of combined NAD^+^ and NAD, but NADH levels do not account for the multiple orders of magnitude difference observed here. Despite these caveats, the relative change in CSF NAD(H) with age is consistent with later reports on brain NAD^+^ levels [93,94].

In 2015, magnetic resonance (MR) was first utilized to measure NAD^+^ and NADH in healthy human brains from a gender-mixed group of 17 people [93]. NAD^+^ was found to be negatively correlated with age, while NADH was found to be positively correlated. Although the participants were separated into age groups, they were not compared directly. Based on their figures, we estimate the overall decrease in brain NAD^+^ levels between the youngest participants (~20 years) and the oldest participants (~60 years) to be in the range of 10–20%.

This study was followed up in 2019 when a similar MR strategy was used by another group to investigate NAD^+^ levels in the brain [94]. This study was comprised of 16 volunteers (12 males, 4 females) between the ages of 26 and 78 and showed a linear relationship between age and NAD^+^, where age explained more than 90% of the total variation in NAD^+^ levels. Based on the parameters of this linear relationship, we estimate a reduction of ~18% in cerebral NAD^+^ between a 25-year-old and a 70-year-old person. The number is not exact, since there were only three participants above the age of 60. Nonetheless, this was consistent with the previously mentioned results on CSF and brain NAD(H) levels [89,93].

In contrast to this, a report deposited on bioRxiv in 2019 used a similar MR approach to evaluate the brain of young (median age 21 ± 4, *n =* 12) and aged (median age 69 ± 4, *n =* 11) humans [95]. The results showed no differences in either NAD^+^ or NADH between the age groups; however, it should be mentioned that these data were removed from the manuscript before publication in *Cell Reports* later that same year [96], and thus have not been peer-reviewed.

Similar to the brain and CSF, blood has also been tested for an association between age and NAD^+^ content by a few independent research groups. As mentioned above, the first investigation in this regard did not directly reveal any changes in plasma NAD(H), despite a correlation with CSF NAD(H) and an effect of age on the latter [89], but this missing effect on plasma has since been challenged. In 2016, in a gender-mixed group of 30 people (15 young < 32 years, 15 elderly > 75 years), it was demonstrated that NAD^+^ was reduced in the elderly group for both whole blood and plasma, but not for red blood cells [97]. Unfortunately, the magnitude of the change in plasma NAD^+^ was not disclosed, since the authors did not provide specific data from plasma or red blood cells aside from *p*-values. They did provide a figure for whole blood, from which we estimated a reduction in NAD^+^ levels of the elderly group of ~15–20%. Intriguingly, this puts the NAD^+^ reduction in the same range as that observed for the brain and CSF. These separate observations could indicate a connection between NAD^+^ levels in plasma/blood, CSF and brain, but aside from a tenuous correlation between plasma and CSF NAD^+^ levels [89], this speculation remains unsupported. Furthermore, whether this level of decline is functionally relevant has been deliberated [89], but no further data have yet been reported to verify or disprove this idea.

A later study reported a negative correlation between plasma NAD^+^ levels and aging in a gender-mixed group [90]. When participants were grouped based on age, a ~80–90% reduction in NAD^+^ was observed between young (20–40 years, *n =* 9) and elderly (≥60 years, *n =* 10) according to our best estimate from reading the published figure. This represents the largest age-related change in human NAD^+^ levels by a large margin and, if factual, means that elderly people have remarkably low plasma NAD^+^ levels. This leaves in total three reports on plasma NAD^+^ and aging, which together present very discrepant extremes ranging from no effect of aging to the almost complete removal of NAD^+^ with age.

Reports on age-related changes in NAD^+^ levels in tissues other than blood and brain are more anecdotal. In 2016, NAD^+^ levels were investigated in livers from a gender-mixed group of middle-aged (<45 years, *n =* 6) and elderly (>60 years, *n =* 6) patients [98]. It is important to mention that all these patients were, at the time, undergoing surgery for hepatocellular carcinoma, although all samples were isolated from non-pathological parts of the liver. The researchers found a ~30% reduction in NAD^+^ between the middle-aged and elderly groups, which was supported by a ~50% reduction in NAMPT [14]. Likewise, our group reported a negative correlation between NAMPT and aged skeletal muscle, resulting in a ~35% reduction in NAMPT between the ages of 20 and 70 years [99], while in adipose tissue, NAMPT levels were unaffected by age. We were, unfortunately, unable to include measurements of NAD^+^ due to a lack of sample material.

There are no reports on the effect of age on actual NAD^+^ levels in human adipose tissue, and only a single report on skeletal muscle. A comparison between NAD(H) levels in calf muscle from young (median age 21 ± 4, *n =* 16) and aged (median age 69 ± 4, *n =* 11) people did not reveal any significant difference [95]. However, similarly to their data on brain NAD^+^ levels, these data were removed from the manuscript prior to peer-reviewed publication [96].

Lastly, two studies of NAD^+^ and aging were performed on cells isolated from humans. In one of the studies, monocyte-derived macrophages from young (≤35 years) and old (≥65 years) people displayed a reduced NAD^+^ level in the older group [9]. Both groups acted as controls in a larger experiment, and they had therefore been transfected with an empty vector prior to measurement of NAD^+^. The other study investigated human mesenchymal stem cells and showed a reduction in NAD^+^ with in vitro aging of the cells [79]. The relevance of this to aging of human tissues is unknown.

Importantly, all of the abovementioned human experiments are cross-sectional studies and had relatively few participants. We can, to some extent, derive an understanding from such studies, but to truly investigate NAD^+^ metabolism in healthy human aging, we need to investigate a much larger group of individuals over the span of many years in a longitudinal study.

Finally, another factor that is largely ignored in the literature when it comes to human NAD^+^ and aging is cellular compartmentalization. We know from animal studies that mitochondrial, nuclear and cytoplasmic NAD^+^ levels are all regulated individually [100]. Therefore, it is essential to investigate the way NAD^+^ levels behave during human aging in these various compartments.

## 3. Conclusions and Perspectives

There are remarkably few studies that assess NAD^+^ levels with aging (Table 1). This is true for most of the commonly used model organisms as well as for humans. Moreover, even within specific tissues, there are discrepancies in the literature, and many tissues in multiple organisms have only been investigated by a single research group or not at all. Thus, there is considerable disagreement between what the field assumes to know on the topic of NAD^+^ in aging and what is scientifically supported. This poor-founded perpetuation of the idea that NAD^+^ levels universally decrease with age is misleading, and it may lead to the loss of important nuances in our collective understanding of NAD metabolism. There is a need for longitudinal studies investigating the way NAD^+^ levels behave in various tissues during aging in various model organisms, and much larger cross-sectional studies in humans are required to address this specific question.

While we believe that we have highlighted all studies reporting lower NAD^+^ levels with aging in the various species, we may have missed relevant studies reporting no change in NAD^+^ with age. This potential unintended selection bias against papers that report no changes in NAD^+^ could be a consequence of general lower motivation to mention or even include such data in a manuscript. Thus, it is likely that even larger discrepancies exist than we are currently aware of, and we therefore encourage the field to push for transparency to gain clarity over this important matter. 

Finally, although NAD^+^ levels may be a suitable readout to gauge the health status of the cell, it is also clear that different NAD-derived metabolites play different roles in this regard. The use of quantitative mass spectrometry-based metabolomics to quantify specific NAD metabolites, which is becoming more commonly used in the field, will provide a better foundation for studying the impact of NAD in relation to aging and disease. Moreover, it should eliminate analytical variability that may account for some of the differences between results obtained by laboratories around the world. In addition, we should not lose sight of the fact that a significant proportion of NAD(P)^+^ is bound as a prosthetic groups of oxidoreductases in the cell, and that there are large differences in the redox state (e.g., NAD^+^/NADH ratio) of different cellular compartments [101]. Thus, future efforts to study the NAD metabolome should not only be directed towards accurate measurements of bound and freely available NAD metabolites, but should also define compartmentalized redox states in response to aging. In turn, this should allow for better pharmacological targeting of relevant tissues/cell types to promote metabolic health as we age.

## Figures and Tables

**Figure 1 nutrients-14-00101-f001:**
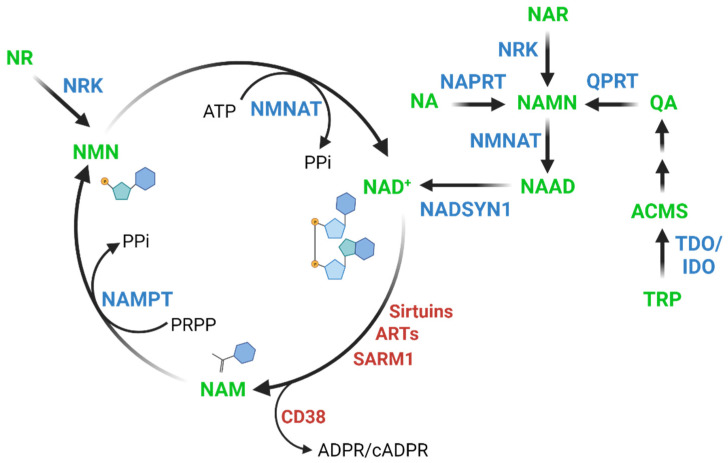
NAD^+^ biosynthesis pathways in mammalian cells. ACMS: 2-amino-3-carboxymuconate, NA: nicotinic acid, NAD^+^: nicotinamide adenine dinucleotide, NAM: nicotinamide, NAMN: nicotinic acid mononucleotide, NAR: nicotinic acid riboside, NMN: nicotinamide mononucleotide, NAAD: nicotinic acid adenine dinucleotide, NR: nicotinamide riboside, QA: quinolinic acid, TRP: tryptophan, IDO: indoleamine-2,3-dioxygenase, NADSYN1: NAD synthetase, NAMPT: nicotinamide phosphoribosyltransferase, NAPRT: nicotinic acid phosphoribosyltransferase, NMNAT: nicotinamide mononucleotide adenylyl transferase, NRK: NR Kinase, QPRT: quinolinic acid phosphoribosyltransferase, TDO: tryptophan-2,3-dioxygenase, ARTs: ADP-ribosyltransferases, CD38: ADP-ribosyl cyclase/cyclic ADP-ribose hydrolase 1, SARM1: NAD^+^ hydroxylase SARM1, ATP: adenosine triphosphate, PPi: inorganic pyrophosphate, PRPP: 5-phosphoribosyl-1-pyrophosphate.

**Figure 2 nutrients-14-00101-f002:**
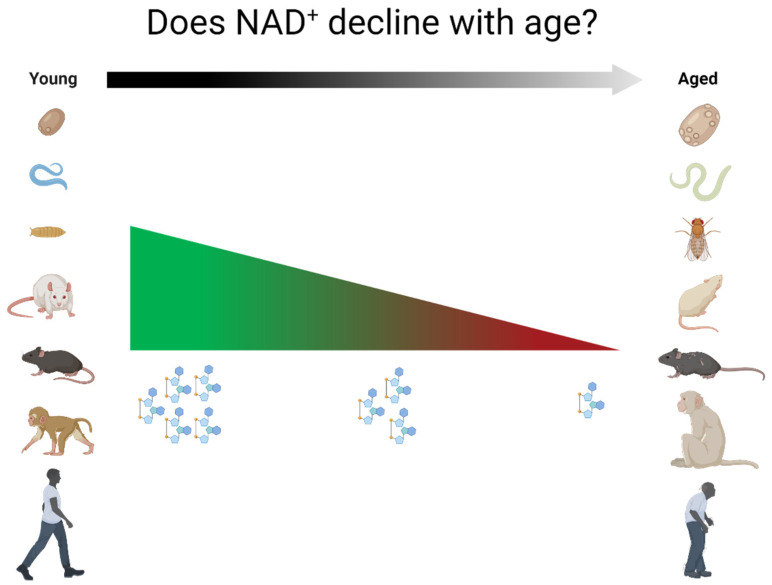
This review discusses the scientific robustness of the claim that NAD^+^ levels decline with age.

**Table 1 nutrients-14-00101-t001:** Overview of studies reporting on NAD^+^ levels with aging.

Species	NAD^+^ Decline with Age	Tissue	Age Comparison	Gender/Sex	Reference # in Manuscript
Yeast	No	n/a	Replicative ages of 0–1 vs. 7–8		[55]
Yeast	No	n/a	Replicative ages of 0 vs. 16		[56]
Yeast	Yes	n/a	Switch to senescent stage		[57]
Yeast	Yes	n/a	Cells grown under increasing optical densities		[50]
*C. elegans*	Yes	Whole worm	Day 1 vs. 8		[58]
*C. elegans*	Yes	Whole worm	Day 1 vs. 17		[59]
Rat	Yes	Heart, lung, liver and kidney	3- vs. 12- vs. 24-month-old female Wistar	Female	[66]
Rat	Yes	Hippocampus, cortex, cerebellum and brainstem	3- vs. 12- vs. 24-month-old female Wistar	Female	[67]
Rat	Yes	Bone Marrow Mesenchymal Stem Cells	Isolated cells from 1–2 vs. 15–18 months-old male Wistar	Male	[68]
Rat	Yes	Bone Marrow Mesenchymal Stem Cells	Senescence-induced cells from 1–2-month-old vs. controls	Male	[69]
Mouse	Yes	Pancreas, adipose tissue, skeletal muscle	3–6 vs. 25–31-month-old	Both sexes	[40]
Mouse	No	Liver	3–6 vs. 25–31-month-old	Both sexes	[40]
Mouse	No	Liver	6- vs. 12- vs. 24- vs. 48-week-old C57BL/6J on HFD or chow	Male	[70]
Mouse	No	Liver	8-,14-,27- vs.55-,110-week-old	Female	[71]
Mouse	Yes	Liver, adipose tissue, spleen, skeletal muscle	5-, 12-, 18-, 24- vs. 32-month-old	Male	[14]
Mouse	Yes	Liver, skeletal muscle	6- vs. 24-month-old	Male	[58]
Mouse	Yes	Skeletal muscle (gastrocnemius)	4- vs. 24-month-old	Male	[73]
Mouse	Yes	Gastrocnemius	6- vs. 22- and 30-month-old	Not specified	[72]
Mouse	Yes	Isolated muscle stem cells	Cells from 1- vs. 22–24-month-old	Male	[5]
Mouse	Yes	Hippocampus	1 vs. 3–4 or 3–4 vs. 6-month-old. No differences: 6-vs. 10–12-month-old	Not specified	[74]
Mouse	Yes	Hippocampus	2-vs.19-month-old. No differences: 2- vs. 7- or 7- vs. 19-month-old	Both sexes	[75]
Mouse	Yes	Dentate gyrus	10- vs. 20–30-week-old	Male	[76]
Mouse	No	whole-brain tissue	2- vs. 8-month-old	Male	[77]
Mouse	No	Cerebellum	4- vs. 16-month-old	Male	[78]
Mouse	Yes	Tail-tip fibroblast	2-vs. 22-month-old	Not specified	[79]
Mouse	Yes	primary peritoneal macrophages	3- vs. 16–20-month-old	Not specified	[9]
Mouse	Yes	BAT, rWAT, iWAT, jejunum, quadriceps, gastrocnemius, soleus, liver, kidney, and descending colon	3- vs. 25-month-old	Male	[80]
Mouse	No	Heart, brain, spleen, pancreas, lungs, proximal colon, duodenum, ileum, gWAT, cecum, stomach	3- vs. 25-month-old	Male	[80]
Human	Yes	Human mesenchymal stem cells	Cells were aged *in vitro*		[79]
Human	Yes	Pelvic skin sample	Spanning 0–77 years of age	Both sexes; children were male	[84]
Human	Yes	Cerebrospinal fluid	Young (<45 years) vs. elderly groups (>45 years)	Both sexes	[89]
Human	No	Plasma	Young (<45 years) vs. elderly groups (>45 years)	Both sexes	[89]
Human	Yes	Brain	Spanning 21–68 years of age	Both sexes	[93]
Human	Yes	Liver	Young (<45 years) vs. old patients (>60 years) undergoing hepatectomy	Both sexes	[98]
Human	Yes	Whole blood, plasma	Young (<32 years) vs. elderly (>75 years) groups	Both sexes	[97]
Human	No	Red blood cells	Young (<32 years) vs. elderly (>75 years) groups	Both sexes	[97]
Human	No	Brain, calf muscle	Young (21 ± 4 years) vs. Elderly 69 ± 4 years) groups	Both sexes	[95]
Human	Yes	Brain	Spanning 26–78 years of age	Both sexes	[94]
Human	Yes	Plasma	Spanning 20–87 years of age	Both sexes	[90]
Human	Yes	Monocyte-derived macrophages	Cells from young (≤35 years) vs. elderly (≥65 years) people	Not specified	[9]

HFD: High-fat diet, BAT: brown adipose tissue, iWAT: inguinal white adipose tissue, rWAT: retroperitoneal white adipose tissue, gWAT: gonadal white adipose tissue.

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
