# Peer review of "Age-Dependent Decline of NAD+—Universal Truth or Confounded Consensus?"

_nutrients, 2021, doi:10.3390/nu14010101_

Round 1
Reviewer 1 Report
The article titled “Age-Dependent Decline of NAD + - Universal Truth or 2 Confusing Consensus? “It deals with the relevance that NAD + levels can have in aging and if these levels go up or down in individuals during aging. The review article is well organized and makes sense from a logical point of view.
- I would just like to propose some aspects to improve the manuscript:
- For better understanding, I recommend that lines 45 to 62 be graphically described with the clues.
- Make a table with the enzymes that use NAD and their compartmentalization.
- Add a section on Drosophila fly experiments.
- I suggest adding a section on physiological actions modifying NAD, it would look great in the review.
- I also suggest that the authors read these two articles to consider their relevance to the manuscript.
Role of NAD+/NADH redox ratio in cell metabolism: A tribute to Helmut Sies and Theodor Bücher and Hans A. Krebs.
Free [NADH]/[NAD+] regulates sirtuin expression.
Author Response
Dear Reviewer,
Thank you for reviewing our manuscript. Below, we will address your concerns.
- As suggested, we have now included a new figure (Figure 1) to provide a graphical overview of the NAD+ biosynthesis routes.
- The overall idea of the manuscript is to highlight papers that show whether or not NAD+ levels change with age. Thus, we have not included a table with enzymes that use NAD+ and their location in the cell as we do not feel that such a table adds significantly to the message of the review. We hope the reviewer agrees with this decision.
- The point about added a paragraph on studies from Drosophila is well taken. We did not do this initially, as we were unable to find any relevant papers. We have now added a paragraph to explain this.
- While we agree with the reviewer that adding information about other "physiological actions" that modify NAD could be very interesting, we also think that such "actions" (e.g., dietary interventions or exercise/fitness level) deserve to be reviewed independently. Moreover, if we were to add this information to this review, and go through all the literature on different dietary interventions across different species, this would take a considerable amount of time, and far longer than the three-day timeline given to us by the editor. Thus, we hope that the reviewer agrees with this decision.
- We read the two papers mentioned by the reviewer. Thank you for pointing to these! They bring about very interesting perspectives. We agree with the reviewer that a 'simple' assessment of total NAD+ levels in the cell per se may confound the interpretation due to the bound fraction, which is not available to the NAD+ consuming enzymes. To highlight this, we have added a paragraph in the conclusion and cited the appropriate reference.
Reviewer 2 Report
Summary of Paper:
The authors provide a comprehensive review of NAD+ with a major focus on age-dependent production. The importance of NAD+ can not be stressed enough as it is has been suggested to be a therapeutic for aging. This reviewer has no major concerns with the article as a if flows well and is exceptionally well written. However, I would recommend that the authors consider making more figures for the manuscript as it is very dense with text and could be a benefit for the reader.
Author Response
Dear Reviewer,
Thank you for reviewing our manuscript. We have revised the manuscript based on the comments from both you and the other reviewer. We have carefully read the text and corrected/adjusted it to improve the language. Moreover, we have included a new figure with an overview of the NAD+ biosynthesis pathways (as also suggested by the other reviewer). We hope that you agree that the current version of the manuscript has improved.
Sincerely,
Jonas Treebak